**Data Availability Statement:** The datasets for this article are not publicly available because of regulations on privacy and confidentiality of patients. Requests to access the datasets should

# What are the risk factors of hospital length of stay in the novel coronavirus pneumonia (COVID-19) patients? A survival analysis in southwest China

Zhuo Wang[1☯], Yuanyuan Liu[2☯], Luyi Wei[2], John S. Ji[3], Yang Liu[1], Runyou Liu[1], Yuxin Zha[1], Xiaoyu Chang[1], Lun Zhang[1], Qian Liu[1], Yu Zhang[1], Jing Zeng[1], Ting Dong[1], Xinyin Xu[1], Lijun Zhou[1], Jun He[1], Ying Deng[1], Bo Zhong[1‡]*, Xianping Wu[1‡]*

**1** Sichuan Center of Disease Control and Prevention, Chengdu, Sichuan, China, **2** Department of Epidemiology and Biostatistics, West China School of Public Health and West China Fourth Hospital, Sichuan University, Chengdu, Sichuan, China, **3** Vanke School of Public Health, Tsinghua University, Beijing, China

☯ These authors contributed equally to this work.
‡ BZ and XW also contributed equally to this work.
* zhongbo1969@163.com (BZ); wwwuxp@163.com (XW)

## Abstract

### Background

The global epidemic of novel coronavirus pneumonia (COVID-19) has resulted in substantial healthcare resource consumption. Since patients' hospital length of stay (LoS) is at stake in the process, an investigation of COVID-19 patients' LoS and its risk factors becomes urgent for a better understanding of regional capabilities to cope with COVID-19 outbreaks.

### Methods

First, we obtained retrospective data of confirmed COVID-19 patients in Sichuan province via National Notifiable Diseases Reporting System (NNDRS) and field surveys, including their demographic, epidemiological, clinical characteristics and LoS. Then we estimated the relationship between LoS and the possibly determinant factors, including demographic characteristics of confirmed patients, individual treatment behavior, local medical resources and hospital grade. The Kaplan-Meier method and the Cox Proportional Hazards Model were applied for single factor and multi-factor survival analysis.

### Results

From January 16, 2020 to March 4, 2020, 538 human cases of COVID-19 infection were laboratory-confirmed, and were hospitalized for treatment, including 271 (50%) patients aged ≥ 45, 285 (53%) males, and 450 patients (84%) with mild symptoms. The median LoS was 19 (interquartile range (IQR): 14–23, range: 3–41) days. Univariate analysis showed that age and clinical grade were strongly related to LoS (P<0.01). Adjusted multivariate analysis showed that the longer LoS was associated with those aged ≥ 45 (Hazard ratio (HR): 0.74,

be directed to Sichuan Center of Disease Control and Prevention Institutional Data Access (approved by the Ethics Committee of Sichuan CDC, scjkwxp@163.com). The original contributions presented in the study are included in the article/supplementary material, further inquiries can be directed to the corresponding authors.

**Funding:** The funding of the project No. 2020-YF05-00296-SN is from Chengdu Science and Technology Bureau. Yuanyuan Liu is the leader of this project, who receives funding award. This funding has nothing to do with any commercial companies. Chengdu Science and Technology Bureau didn't play any role in study design, data collection and analysis, decesion to publish, and preparation of the manuscript. This research was supported by the Grants from Science and Technology Bureau of Sichuan province (COVID-19 science and technology emergency project, No. 2020YFS0015), Sichuan Provincial Leading Group Office for Talent Work (Sichuan provincial "Tianfu ten thousand talents plan" fund in 2018) and China National Natural Science Funding (No.82041033), Science and Technology Department of Chengdu (No. 2020-YF05-00296-SN).

**Competing interests:** The authors have declared that no competing interests exist.

**Abbreviations:** CDC, Center for Disease Control and Prevention; CI, Confidence interval; COVID-19, Novel coronavirus pneumonia; FiO2, Fraction of inspired oxygen; HR, Hazard ratio; IQR, Interquartile range; LoS, Length of stay; NGS, Next-Generation Sequencing; NNDRS, National Notifiable Diseases Reporting System; PAHO, Pan American Health Organization; PaO2, Pressure of oxygen; PHC, Primary healthcare; RR, Respiratory rate; RT-PCR, Reverse-transcriptase polymerase chain reaction; WHO, World Health Organization.

95% confidence interval (CI): 0.60–0.91), admission to provincial hospital (HR: 0.73, 95% CI: 0.54–0.99), and severe illness (HR: 0.66, 95% CI: 0.48–0.90). By contrast, the shorter LoS was linked with residential areas with more than 5.5 healthcare workers per 1,000 population (HR: 1.32, 95% CI: 1.05–1.65). Neither gender factor nor time interval from illness onset to diagnosis showed significant impact on LoS.

## Conclusions

Understanding COVID-19 patients' hospital LoS and its risk factors is critical for governments' efficient allocation of resources in respective regions. In areas with older and more vulnerable population and in want of primary medical resources, early reserving and strengthening of the construction of multi-level medical institutions are strongly suggested to cope with COVID-19 outbreaks.

## Introduction

The first case of novel coronavirus disease-2019 (COVID-19) was admitted to hospital in Wuhan, Hubei province, on December 12, 2019 [1]. On December 31, 2019, China reported for the first time to the World Health Organization (WHO) Country Office an unexplained case of pneumonia found in Wuhan. On January 30, 2020, the COVID-19 outbreak was announced as a public health emergency of international concern. On March 11, 2020, WHO characterized the COVID-19 outbreak as a pandemic [2, 3].

So far, many studies have reported epidemiological, clinical features, molecular and biological mechanisms, as well as prevention and control management of COVID-19 [4–6]. These descriptive studies enabled researchers and policy makers to understand the incubation period and transmutability of SAR-CoV-2 [7, 8]. With the global spread of COVID-19, more and more regions raised the concern that the epidemic has imposed a great burden on health resources. Advance attention and evaluation on resouce consumption was demanded, and insight into COVID-19 patients' hospital length of stay (LoS) and its risk factors could contribute to the proper allocation of medical resources. Eleanor M. Rees et al. carried out systematic analysis and proposed that the bed demand of emergency plan ccould be simulated by LoS, so as to predict the demand of health resources [9]. Knowledge on factors affecting hospital LoS is evolving, and some studies reported that LoS of COVID-19 patients might be affected by the severity of disease, medication, comorbidity and other factors [10–12]. However, most reported risk factors related to LoS were clinical predictors and the role of social determinants were still unexplored.

To fill this gap and support the preparation and allocation of health resources, we assessed the relationship between LoS and several social determinants, namely, demographic characteristics of the confirmed patients, individual treatment behavior, local medical resources and hospital grade. We chose Sichuan, which is near Hubei, as the province investigated for its huge population of over 80 million [13].

## Materials and methods

### Data sources

We applied retrospective data on laboratory-confirmed cases of COVID-19 in Sichuan province of China, which were reported to the Sichuan Center for Disease Control and Prevention (CDC) through the National Notifiable Diseases Reporting System (NNDRS) [14]. In addition to the network of data reported by hospitals at all levels, part of the data came from a field

survey of confirmed cases conducted by the staff of local or provincial CDC and Health Resources Report of Health Commission of Sichuan Province [15].

Demographic, epidemiological, and basic clinical data were obtained by combining these data sources. The information of 538 confirmed patients in Sichuan province from January to March 2020 was analyzed, including their age, sex, place of residence, dates of illness onset, dates of diagnosis, hospital admission, discharge, clinical grade, hospital grade, and health service personnel per 1,000 population. The period of data spanned from January 16, 2020, when the first case was confirmed in Sichuan province, to March 4, 2020.

## Patients

Diagnostic and treatment protocol for Novel Coronavirus Pneumonia (Interim version 7) was released by National Health Commission of the People's Republic of China, which specifies case definitions, diagnosis, differential diagnosis, treatment, laboratory assays and discharge criteria [16]. Confirmed cases were determined on the basis of epidemiological history, clinical manifestations and laboratory examinations. In addition to possible contact with COVID-19 cases and/or having the clinical manifestations of COVID-19 infection, confirmed case also needs to meet one of the following etiological or serological evidences, including changes in nasopharyngeal swabs, sputum, other lower respiratory tract secretions, blood, feces and other specimens:

1. Novel Coronavirus nucleic acid is tested positive by reverse-transcriptase polymerase chain reaction (RT-PCR).

2. Viral gene sequencing by Next-Generation Sequencing (NGS) is highly homologous with the known Novel Coronavirus.

3. Novel Coronavirus specific IgM and IgG antibodies are positive in serum.

4. Novel Coronavirus specific IgG antibody in the serum changed from negative to positive or the novel coronavirus specific IgG antibody is four times or more elevated in the recovery period than in the acute phase.

Clinical grades were divided into two levels: mild and severe illness. The mild type was characterized by mild clinical symptoms, with or without fever, respiratory tract symptoms, and imaging manifestations of pneumonia. For severe type, the characteristics in adults should meet any of the following criteria:

1. Shortness of breath with respiratory rate (RR) is greater than or equal to 30 per minute;

2. Under resting state, finger oxygen saturation is less than or equal to 93%;

3. Arterial partial pressure of oxygen divided by fraction of inspired oxygen (PaO2/FiO2) is less than or equal to 300mmHg.

Children of severe type should meet any of the following criteria:

1. Shortness of breath (< 2 months, RR greater than or equal to 60 per minute; 2–12 months, RR greater than or equal to 50 per minute; 1–5 years old, RR greater than or equal to 40 per minute; over 5 years old, RR greater than or equal to 30 per minute), except for the circumstances of fever and crying;

2. Under resting state, the finger oxygen saturation is less than or equal to 92%;

3. Auxiliary breathing (moaning, flapping of the nose, triple concave sign), cyanosis, intermittent apnea;

4. Lethargy and convulsion;

5. Refuse to eat or feeding difficulties, dehydration sign.

The characteristics of severe type also included lung imaging showing that the lesions obviously progressed more than 50% within 24–48 hours, and even respiratory failure, shock or organ failure. Discharge criteria were: 1) the body temperature returned to normal for more than 3 days; 2) the respiratory symptoms improved significantly; 3) the pulmonary imaging showed a significant improvement in acute exudative lesions; 4) the nucleic acid tests are negative for two consecutive respiratory specimens (sampling interval at least 1 day apart). All four criteria must be met before discharge. Hospital grades were divided into three categories: provincial level, city level and county level, which were managed by different levels of government, and we combine the city and county level into non-provincial levels.

## Ethics approval and consent to participate

The study was approved by the Ethics Committee of Sichuan CDC (SCCDCIRB-2020-006) and the written informed consent was waived because this study is retrospective in nature and falls under the category of emergency medical service.

## Statistical analysis

In this study, the events of interest were discharge status. Figure of 1 represented discharge, while 0 meant no discharge by the end of March 4, 2020 (censoring, treated as incomplete data); and death (there were three deaths in Sichuan province in this period) was also treated as 0. That is, our outcome event was discharge. Survival time was hospital LoS. Survival rate was cumulative probability of hospitalization.

The relationship between patient's age, gender, time interval from illness onset to diagnosis, hospital grade of patients, healthcare workers per 1,000 population in the patient's permanent residence, clinical grade, and hospital LoS was analyzed by survival analysis. Firstly, the Kaplan-Meier method was used to estimate survival rate, which was cumulative probability of hospitalization; and the log-rank test was used to compare survival curves in univariate analysis. Then, the Cox Proportional Hazards Model was used for multi-factor analysis and to determine the potential risk factors related to LoS [17]. Schoenfeld residuals was used to check the proportional hazards assumption. Cox proportional hazards regression model can be written as follows:

$$h(t) = h_0(t)\exp(\beta_1\chi_1 + \beta_2\chi_2 + L + \beta_k\chi_k)$$

where $h(t)$ was instantaneous rate of experiencing discharge status at hospital stay length $t$ for a patient with a set of predictors $x_1, x_2, \ldots, x_k$; $h_0(t)$ was the baseline hazard function; covariates $X$ were the patient's age, gender, time interval from illness onset to diagnosis, hospital grade, healthcare workers per 1,000 population in the patient's permanent residence, clinical grade of patients, and $\beta_1, \beta_2, \ldots, \beta_k$ were the model parameters describing the effect of the predictors on the overall hazard. Briefly, hazard ratio (HR) > 1 indicated an increased probability of hospital discharge if a specific condition was met by a patient. While HR < 1, on the other hand, indicated a decreased probability of hospital discharge, which also meant a longer hospital LoS.

In order to build a more interpretable model, we transformed continuous variables (age, time interval from illness onset to diagnosis, healthcare workers per 1,000 population) and multi-category variable (hospital grade) into binary variables according to the variables' distribution. The main principle of choosing cutoffs for continuous or multi-category variables was to ensure that the number of cases in two groups were comparable, such as selecting the

median age as cutoff. Sensitivity analyses were conducted to evaluate the stability of the results by (1) changing the variable type of risk factors which had been transformed in the main analysis, or (2) excluding the three deaths. Statistical analyses were conducted in R (version 3.6.3), and *P*-values less than 0.05 for parameter estimates were considered statistically significant.

## Results

### The characteristics of patients

The first confirmed case in Sichuan province was hospitalized on January 16, 2020. By March 4, 2020, 538 laboratory-confirmed human cases of COVID-19 infection have been hospitalized for treatment in Sichuan province. The flow diagram of diagnostic protocol for COVID-19 is showed in Fig 1. Based on comparable principle, the age groups were divided into <45 years old and ≥45 years old, the time interval groups from onset to diagnosis were divided into <5 days and ≥5 days, the number groups of healthcare workers per 1,000 population were divided into <5.5 and ≥5.5, and the hospital grade groups were divided into provincial and non-provincial levels. Of those, 285 (53%) were male, 450 (84%) had mild diseases, 301 (83%) were admitted to hospitals below the provincial level, 364 (68%) recovered and discharged, and three (0.6%) died (Table 1).

### The length of stay in the hospital

The median LoS for all confirmed inpatients was 19 (interquartile range (IQR): 14–23, range: 3–41) days, while it was 21 (IQR: 14–24, range: 3–41) days for those aged 45 years old and above, 18 (IQR: 13–22, range: 3–41) days for people under 45 years old, 18 (IQR: 13–24, range: 3–41) days for patients with at least a 5-day interval from illness onset to visit hospital, 19 (IQR: 14–23, range: 3–41) days for patients who were admitted to hospitals below the provincial level, 18 (IQR: 12–23, range: 3–41) days for patients living in areas with a medical resource density greater than 5.5 healthcare workers per 1,000 population, and 19 (IQR: 14–23, range: 3–39) days for mild cases (Table 1 and Fig 2).

### Survival analysis

Kaplan-Meier curves and log-rank test comparisons indicated statistically significant differences in the duration of hospital stay by age group and clinical grade (P<0.01, Fig 2). The median LoS in the group ≤ 45 years old was 3 days shorter than that in the group ≥ 45 years old, while it was 4 days shorter in mild patients than in severe patients. The probability of discharge over time among mild younger patients was higher than that of the severe older patients.

Fig 3 described the results of multivariate Cox Proportional Hazards Model and identified the factors associated with hospital LoS for COVID-19 with the lowest AIC and BIC. As we mentioned in the section of "Statistical analysis", HR less than one indicated that the probability of discharge was reduced and the risk of prolonged hospitalization was increased. The results showed that patients aged ≥ 45 years old (HR: 0.74, 95% confidence interval (CI): 0.60–0.91, P = 0.005), admitted to provincial hospital (HR: 0.73, 95% CI: 0.54–0.99, P = 0.040), and having severe illness (HR: 0.66, 95% CI: 0.48–0.90, P = 0.008) had longer LoS; while living in areas with more than 5.5 healthcare workers per 1,000 population (HR: 1.32, 95% CI: 1.05–1.65, P = 0.016) had shorter LoS. The HR of age groups indicated that compared to patients aged < 45 years, the patients those ≥ 45 years old had an increase of 26% in the risk of continued hospital treatment. Similarly, compared with the reference groups, the patients who admitted to provincial hospital and with severe illness had increased risk of 27% and 34% in

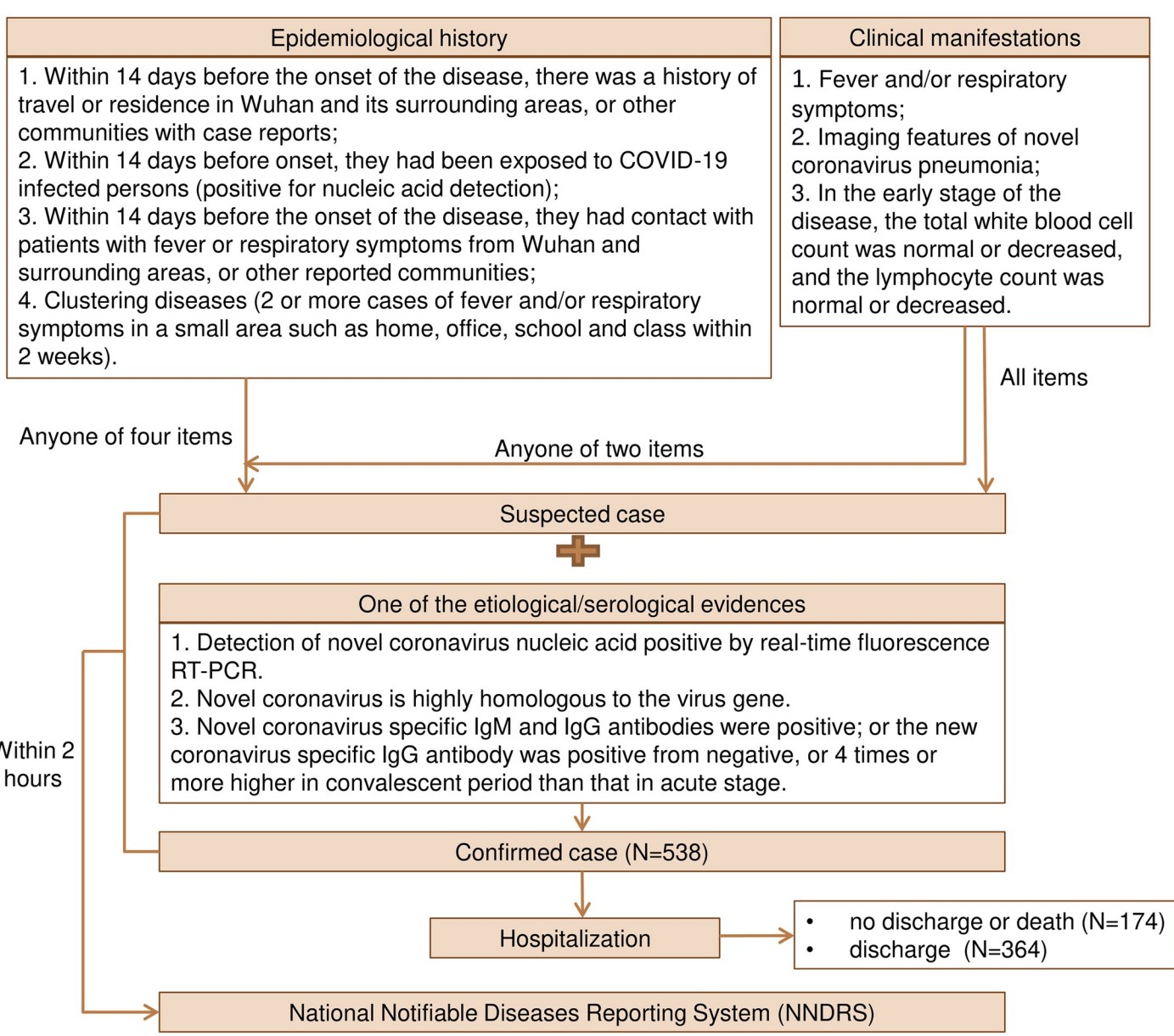

| Epidemiological history | Clinical manifestations |
| --- | --- |
| 1. Within 14 days before the onset of the disease, there was a history of travel or residence in Wuhan and its surrounding areas, or other communities with case reports;<br>2. Within 14 days before onset, they had been exposed to COVID-19 infected persons (positive for nucleic acid detection);<br>3. Within 14 days before the onset of the disease, they had contact with patients with fever or respiratory symptoms from Wuhan and surrounding areas, or other reported communities;<br>4. Clustering diseases (2 or more cases of fever and/or respiratory symptoms in a small area such as home, office, school and class within 2 weeks). | 1. Fever and/or respiratory symptoms;<br>2. Imaging features of novel coronavirus pneumonia;<br>3. In the early stage of the disease, the total white blood cell count was normal or decreased, and the lymphocyte count was normal or decreased. |

Anyone of four items | Anyone of two items | All items

**Suspected case**

**One of the etiological/serological evidences**

1. Detection of novel coronavirus nucleic acid positive by real-time fluorescence RT-PCR.
2. Novel coronavirus is highly homologous to the virus gene.
3. Novel coronavirus specific IgM and IgG antibodies were positive; or the new coronavirus specific IgG antibody was positive from negative, or 4 times or more higher in convalescent period than that in acute stage.

**Confirmed case (N=538)**

Within 2 hours

**Hospitalization**

- no discharge or death (N=174)
- discharge (N=364)

**National Notifiable Diseases Reporting System (NNDRS)**

**Fig 1. Flow diagram of diagnostic protocol for novel coronavirus pneumonia (COVID-19).**

the continued hospital treatment, respectively. Meanwhile, compared to patients living in areas with more than 5.5 healthcare workers per 1,000 population, those living in areas with less than 5.5 healthcare workers per 1,000 population had an increase of 32% in the risk of longer hospitalization.

Alternatively, we also considered age and density of healthcare workers as continuous variables and used stepwise-selected Cox model to calculate HR. The sensitivity analysis results showed the relationships between these four variables and LoS were still statistically different (see S1 Fig). The sensitivity analysis results also showed that LoS was not affected after excluding three deaths (see S2 Fig).

**Table 1. Characteristics and LoS of 538 COVID-19 confirmed cases admitted to hospital.**

| Group | Discharged[a] | Not discharged / Died | Total | Median estimate of LoS [b] | 95% Confidence interval | | IQR of LoS [c] | Range of LoS |
|---|---|---|---|---|---|---|---|---|
| | | | | | Lower limit | Upper limit | | |
| Total | 364 | 174 | 538 | 19 | 18 | 21 | 14–23 | 3–41 |
| Age (years) | | | | | | | | |
| <45 | 200 (55%) | 67 (39%) | 267 (50%) | 18 | 17 | 19 | 13–22 | 3–41 |
| ≥45 | 164 (45%) | 107 (61%) | 271 (50%) | 21 | 19 | 23 | 14–24 | 3–41 |
| Gender | | | | | | | | |
| Male | 192 (53%) | 93 (53%) | 285 (53%) | 19 | 18 | 21 | 14–24 | 3–41 |
| Female | 172 (47%) | 81 (47%) | 253 (47%) | 19 | 18 | 21 | 14–23 | 3–37 |
| Time interval from illness onset to diagnosis (days) | | | | | | | | |
| <5 | 214 (59%) | 105 (60%) | 319 (59%) | 19 | 19 | 21 | 14–23 | 3–39 |
| ≥5 | 150 (41%) | 69 (40%) | 219 (47%) | 18 | 17 | 21 | 13–24 | 3–41 |
| Hospital grade | | | | | | | | |
| Provincial | 63 (17%) | 56 (32%) | 119 (22%) | 20 | 15 | NA | 10–24 | 3–41 |
| Non- provincial | 301 (83%) | 118 (68%) | 419 (78%) | 19 | 18 | 20 | 14–23 | 3–41 |
| Healthcare workers per 1,000 population | | | | | | | | |
| <5.5 | 174 (48%) | 96 (55%) | 270 (50%) | 20 | 19 | 22 | 15–23 | 5–39 |
| ≥5.5 | 190 (52%) | 78 (45%) | 268 (50%) | 18 | 17 | 21 | 12–23 | 3–41 |
| Clinical grade | | | | | | | | |
| Mild illness | 315 (87%) | 135 (78%) | 450 (84%) | 19 | 17 | 19 | 14–23 | 3–39 |
| Severe illness | 49 (13%) | 39 (22%) | 88 (16%) | 23 | 21 | 31 | 15–26 | 3–41 |

[a] Data are n (%).

[b] The estimate was unadjusted.

[c] IQR: interquartile range.

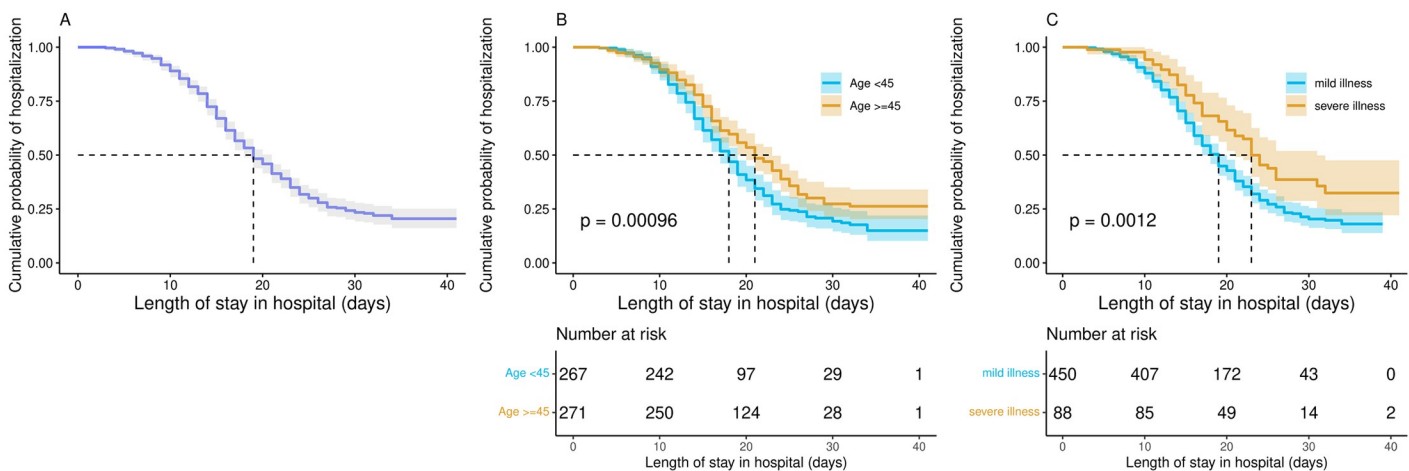

**Fig 2. Cumulative probability of hospitalization Kaplan–Meier survival curves with at-risk tables for COVID-19 confirmed patients.** Shading shows 95% CIs. 538 COVID-19 confirmed patients of all ages (A), 267 patients younger than 45 years and 271 patients at least 45 years (B), 450 mild cases and 88 severe cases (C).

## Discussion

Novel coronavirus pneumonia has become a major public health event owing to its rapid transmission, large-scale dissemination and the heavy load on medical resources. Therefore, it'

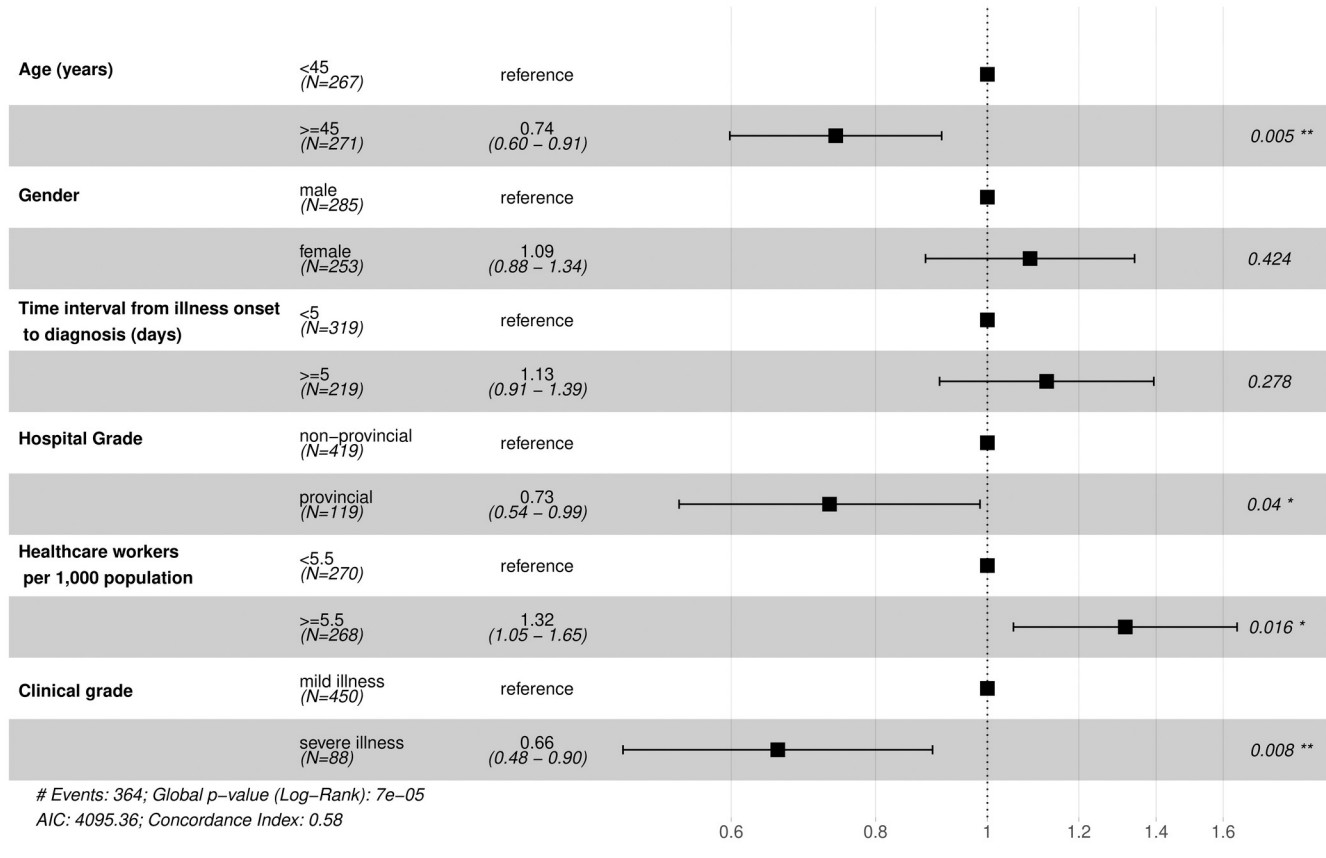

**Fig 3. Risk factors for LoS of COVID-19 confirmed patients.**

s strongly suggested to prepare for risk prevention and control in advance according to the local demographics features and the current situations of medical resources. As of September 2020, the number of confirmed cases of COVID-19 in Sichuan was less than 700, and more than 80% of the confirmed cases occurred before March. In order to highlight the most serious epidemic period in Sichuan province, we showed LoS of COVID-19 infection patients diagnosed in Sichuan from January 16, 2020 to March 4th. At that time, almost all the cases, from imported ones to local infections in Sichuan province, were concentrated in this period. We modeled demographic characteristics, treatment behaviors, clinical characteristics of hospitalized confirmed patients and local medical resources, so as to inform prospective risk assessment for other areas. Based on the hospitalization data of confirmed patients in Sichuan province, we found that patients ≥ 45 years old, having severe illness, living in areas with fewer healthcare workers per 1,000 people and being admitted to higher hospital grade had longer lengths of hospitalization, while the factors of gender and time interval from onset to visit the hospital had no effect on hospital LoS. That is to say, the age composition of patients, the proportion of severe illness, the density of health service, and the hospital grade of patients were important factors affecting the length of hospital stay. In contrast, whether patients see a doctor in time did not seem to affect LoS, as is shown by the single factor and multi-factor survival analyses; there was no significant difference in the LoS between the two groups whose time interval from onset to diagnosis were within 5 days and more than 5 days.

Hospital LoS due to COVID-19 has been reported in several studies in China. A systematic review identified 52 studies and reported that summary median hospital LoS was 14 (IQR:10–

19, range: 4–53) days for China and 5 (IQR:3–9, range: 4–21) days outside of China [9]. Sichuan province, as a populous and frequently travelled region neighboring Hubei, had some prevention and control measures worth sharing with other regions [18]. On January 21, 2020, the Sichuan provincial government began to carry out a series of epidemic prevention measures, including conducting epidemic prevention training for primary healthcare workers, formulating new plans to allocate resources, preparing medical resources that would be used for at least until May, requiring patients with mild illness or suspected patients to select hospitals as close as possible to avoid cross-infection, etc [19]. After the completion of these steps, the province-wide "first level emergency response" was launched on January 24, 2020, demanding the prohibition of any form of group gathering activities and the avoidance of public panic [20]. This chain of precautions may also account for why the confirmed patients' median LoS (19 days) in Sichuan province was longer than the national average, which was 10–14 days, so that patients can get longer treatment and care in the hospital [1].

In China, the majority of cases were classified as mild (81%). The overall mortality rate was estimated to be 2.3% [21, 22]. Although the mortality rate of COVID-19 was lower than that of SARS (10%) and MERS (34%), the total deaths number of COVID-19 globally exceeded SARS and MERS [23–25]. We posited that insufficient medical resources or unreasonable allocation of medical resources may be one of the reasons for the excessive number of deaths. For example, many patients with mild illness may occupy many medical resources. Massonnaud et al. (2020) assessed COVID-19's impact on healthcare resources for each French metropolitan region based on the average LoS of 15 days, and found that even in the best case scenario, the French healthcare system would be at the brink of crumbling [26]. Most reported risk factors related to Hospital LoS were older age, severe pneumonia [10, 27]. Pham and his colleagues reported that age, residence and sources of contamination were significantly associated with longer duration of hospitalization in Vietnam [28]. Our paper showed the necessity of establishing a multi-level medical system and strengthening primary healthcare (PHC). In Sichuan province, 78% (419/538) of the patients were admitted to non-provincial hospitals. Single factor analysis showed that the levels of hospitals had no effect on the length of inpatient treatment. After controlling the covariates such as patient age, clinical severity and regional medical service density, we found that the hospital grade had a weak association with the LoS, which is mainly attributed to the clinical severity of patients. Patients in areas with more than 5.5 healthcare workers per 1,000 population had shorter hospital stays. Lower level hospitals and PHC may reduce the hospitalization rate of COVID-19 patients through the following ways. Firstly, PHC doctors can provide timely acute, chronic and preventive healthcare for patients; secondly, PHC can care for more people, including vulnerable groups in rural areas; thirdly, multi-level hospitals can divert mild patients and reduce the pressure of diagnosis and treatment in high-level hospitals; fourth, PHC can strengthen public health information to help patients manage at home [29]. In addition, PHC doctors also play an important role in the dissemination of health knowledge. It has been reported that the severity of the epidemic in the United States was related to PHC shortage [30]. Sarah Mitchell et. also reported that PHC also played an important role in palliative treatment and response to the epidemic [31, 32]. In recent years, China has been strengthening the construction of lower-level medical systems and the monitoring of major diseases and health hazards, which deserve further implementation [33].

It is our joint and sincere hope that all patients shall receive timely treatment. However, in the cases where there is a lack of medical resources or where patients mistaken COVID-19 for influenza, treatment may be delayed. Although influenza is more contagious than COVID-19 in transmission, it is less severe, and the hospitalized patients are mostly the elderly and children and it's LoS is generally 5 days [34, 35]. The results of this study showed that the interval

length of time from symptoms to diagnosis did not affect the LoS, so quarantine was more important. We suggested that in areas with insufficient medical resources and shortage of inpatient beds, patients under 45 years old with mild illness could be treated and quarantined at home under the guidance of doctors, which can avoid cross infection and crowded consumption of medical resources, as long as safe medical care was available at home and timely communication between doctors and patients was possible. While encouraging quarantine, best-practices should be considered. The following suggestions can be considered: placing patients in an isolated single room with good ventilation and away from visitors; changing of clothing, bed linen, and utensils for personal care every day if possible entering and leaving the room with protective measures (protective gown, gloves), etc [36–39]. In addition to individual recommendations, authoritative agencies such as WHO, Pan American Health Organization (PAHO) have also issued Hospital Readiness Checklist for COVID-19 for hospitals to quickly assess the coordination of health facilities. For example, for isolation, it was proposed to provide triage space available in the emergency area, with isolation measures for suspected and confirmed cases; identify, sign, and equip areas for medical care of suspected and confirmed cases in secure and isolated conditions; review, update and test hospital procedures for receiving and transferring patients to authorized quarantine areas and other diagnostic and treatment support services [40, 41]. As far as we know, this is the first time in Sichuan province to explore the situation of LoS and its determinant factors in patients with COVID-19 in the most severe period combined with multi-source data. Previously, some researchers proposed the need to identify groups that may have poor outcomes [42]. This paper hoped, through the study of LoS, to provide reference for these needs. Compared with European countries, Sichuan province has the similar size of area as Spain (486 thousand square meters), more population than Britain and Italy (more than 80 million), similar population density as Italy (171 people per square kilometer), similar proportion of elderly population as Ireland (the proportion of 65 years old and above is more than 14%), lower economic level (GDP per capita less than 50 thousand RMB) [43], and less confirmed cases. Based on our paper, we believe that Sichuan story can offer other epidemic areas more practical value for the policy-making and measures.

Inevitably, this study contains several limitations. First, there may be other affecting factors of LoS left out of the analysis, such as whether the diagnosed patient had comorbidity. Second, this study could have further divided the confirmed cases of COVID-19 into imported cases or local cases to obtain more detailed inpatient characteristics. To advance the field, further studies may explore other pathways to reveal the impact of other social and individual factors on LoS of COVID-19.

## Conclusions

Understanding the length of stay of COVID-19 patients is critical for the allocation of regional medical resources. Based on our finding, the government can assess and prepare medical resources in advance in regions with vulnerable and older populations and limited primary medical resources Moreover, we also need to strengthen the construction of multi-level medical institutions to deal with public health emergencies and medical resources occupation.

## Supporting information

**S1 Fig. Sensitivity analysis of hazard ratio (HR) of changing the variable type of risk factors.**
(TIFF)

**S2 Fig. Sensitivity analysis of risk factors' hazard ratio (HR) of excluding the three deaths.** (TIFF)

## Acknowledgments

We thank all staff in CDCs at all levels for their contributions to fight against novel coronavirus epidemic.

## Author Contributions

**Conceptualization:** Zhuo Wang, Yuanyuan Liu.

**Data curation:** Zhuo Wang, Luyi Wei, Runyou Liu, Yuxin Zha.

**Funding acquisition:** Zhuo Wang, Yuanyuan Liu, Bo Zhong, Xianping Wu.

**Methodology:** Zhuo Wang, Yuanyuan Liu, John S. Ji, Bo Zhong, Xianping Wu.

**Supervision:** Bo Zhong, Xianping Wu.

**Writing – original draft:** Zhuo Wang, Yuanyuan Liu, Runyou Liu, Yuxin Zha.

**Writing – review & editing:** Luyi Wei, Yang Liu, Xiaoyu Chang, Lun Zhang, Qian Liu, Yu Zhang, Jing Zeng, Ting Dong, Xinyin Xu, Lijun Zhou, Jun He, Ying Deng, Xianping Wu.

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
