## [Decision Letter · Decision Letter 0]

8 Sep 2021

PONE-D-21-21225What are the risk factors of hospital length of stay in the novel coronavirus (COVID-19) pneumonia patients: a prospective study in southwest ChinaPLOS ONE

Dear Dr. Zhong,

Thank you for submitting your manuscript to PLOS ONE. After careful consideration, we feel that it has merit but does not fully meet PLOS ONE’s publication criteria as it currently stands. Therefore, we invite you to submit a revised version of the manuscript that addresses the points raised during the review process.

We look forward to receiving your revised manuscript.

Kind regards,

Yang Yang, Ph.D

Academic Editor

PLOS ONE

Journal Requirements:

Additional Editor Comments (if provided):

Please address the reviewers' comments, most of which I think are reasonable. Please also consider the following points:

1. You incorporated all 538 patients, and the LoS of the 3 fatal cases were considered right-censored. A more rigorous statistical analysis would treat discharge (recovery) and death as two competing risks. However, I don't think a competing-risk analysis is necessary given such as small number of deaths. I'd recommend a sensitivity analysis excluding the three deaths and compare the results.

2. You used the number of healthcare workers per 1000 population as a predictor. Sometimes it is referred to as the number of healthcare providers per 1000 in the manuscript. Healthcare workers and healthcare providers are different concepts. The former is individual-based, which include physicians, nurses and others. The latter is more institute-based, referring to clinics, hospitals, etc. Pick one term that fits your data and use it throughout the manuscript.

Reviewers' comments:

Reviewer's Responses to Questions

**Comments to the Author**

1. Is the manuscript technically sound, and do the data support the conclusions?

Reviewer #1: Yes

Reviewer #2: Yes

2. Has the statistical analysis been performed appropriately and rigorously? 

Reviewer #1: Yes

Reviewer #2: I Don't Know

3. Have the authors made all data underlying the findings in their manuscript fully available?

Reviewer #1: No

Reviewer #2: No

4. Is the manuscript presented in an intelligible fashion and written in standard English?

Reviewer #1: No

Reviewer #2: Yes

5. Review Comments to the Author

Reviewer #1: This manuscript aims to find the effective demographical covariates on the length of hospitalization for the COVID-19 patients. The overall presentations of the manuscript, including the background, data explanation, and discussion, are satisfactory. However, some points can improve the manuscript in terms of presentation and methodology. Here are my comments:

1. The whole manuscript needs revision for English proficiency. For instance, the authors should revise the background and conclusion sections at the start of the manuscript.

2. It would be better if the authors use the full word instead of the abbreviation when using it for the first time. For instance, in the results section of page 2, define HR; otherwise, the readers need to look for the abbreviation list when they start to read.

3. In figure 2, the Kaplan-Meier plots show the probability of hospitalization based on the length of the hospital stay, which may be misleading. Once a person stayed in the hospital, the length of stay does not affect hospitalization probability unless the authors' purpose was “the discharge probability”.

4. The authors need to describe the plots with more details in the “results of survival analysis”.

5. It would be better if the authors could provide evidence that the Cox Proportional hazard model’s assumptions are satisfied by their data.

In general, the manuscript has the potential of providing scientific research supported by data analysis. But it needs more revisions before publishing.

Reviewer #2: The study looks at important public health issue.

While other researchers in China on the similar issue have focused more on clinical predictors of Length of Stay, the current paper looks at social determinants, and thus adds value to current understanding of the subject.

1. The title of the manuscript presents this as a prospective study. However, the study was performed on the data tat was already collected through NNDRS, and filed survey. The design qualifies this as retrospective database study.

2. Authors mention that all subjects gave their informed consent for inclusion before they participated. It needs further elaboration. How did subjects gave their consent if the collected data through NNDRS was obtained for the study purposes.

3. HR below 1 intuitively mean less risk of developing an outcome/ event. Here more appropriate event should be related to Length of stay, rather than discharge. I urge authors to use HR more intuitively.

4. The manuscript mentions that the number of confirmed cases of COVID-19 was less than

700 as of September, 2020, and more than 80% of the confirmed cases occurred before

March (meaning that about 560 cases had occurred before March). Of these, the study looks at 538 hospitalized patients. Does it mean that >95% of confirmed cases were being hospitalized?

5. No need to mention few results. It makes the manuscript appear less thoughtful. Authors need to review which results need to be presented carefully (Under section 3.1 authors say that 271 (50%) of the 538 individuals who had to be admitted to hospital were aged at least 45 years. This is not needed because the cut-off of 45 years was chosen as it was median. Similarly, under section 3.2 no need to mention that the LoS was 19 days for both men and women)

6. Surprisingly, authors do not discuss any of the multiple papers published based on data from China looking at the factors affecting LoS in COVID patients. It is important to put the finding of this paper in the context of what other researchers have found in China.

7. The visual clarity of figures is very bad making it impossible to comment.

6. PLOS authors have the option to publish the peer review history of their article (what does this mean?). If published, this will include your full peer review and any attached files.

Reviewer #1: No

Reviewer #2: No

---

## [Author Response · Author response to Decision Letter 0]

11 Oct 2021

Dear editor,

Thank you very much for your constructive comments and suggestions. We have revised the manuscript according to Journal Requirements, and addressed the editor’s and reviewers’ comments point by point as listed below to improve the quality and clarity of the manuscript. We hope the revision is now acceptable.

Additional Editor Comments:

1. You incorporated all 538 patients, and the LoS of the 3 fatal cases were considered right-censored. A more rigorous statistical analysis would treat discharge (recovery) and death as two competing risks. However, I don't think a competing-risk analysis is necessary given such as small number of deaths. I'd recommend a sensitivity analysis excluding the three deaths and compare the results.

Response:

We highly appreciate your suggestion. This time we have excluded the three deaths, added sensitivity analysis, and compared the results following your advice. The three deaths appeared to have no effect on the results of this paper and detailed information were added in the third paragraph of Statistical analysis section and Figure S2.

2. You used the number of healthcare workers per 1000 population as a predictor. Sometimes it is referred to as the number of healthcare providers per 1000 in the manuscript. Healthcare workers and healthcare providers are different concepts. The former is individual-based, which include physicians, nurses and others. The latter is more institute-based, referring to clinics, hospitals, etc. Pick one term that fits your data and use it throughout the manuscript.

Response:

Thank you for pointing this out for us. We have checked the full text and replaced “healthcare providers” with “healthcare workers”.

Reviewer #1:

1. The whole manuscript needs revision for English proficiency. For instance, the authors should revise the background and conclusion sections at the start of the manuscript.

Response:

We are truly grateful for your suggestions. With the professional help, we have revised the English expression of the whole manuscript, including the background and conclusion sections.

2. It would be better if the authors use the full word instead of the abbreviation when using it for the first time. For instance, in the results section of page 2, define HR; otherwise, the readers need to look for the abbreviation list when they start to read.

Response:

Thanks for pointing this out for us. We have checked the full text and reconfirmed that we used the full words with abbreviations for the first time. For example, we have modified HR to “Hazard ratio (HR)” in Abstract and Statistical analysis section where it was used for the first time.

3. In figure 2, the Kaplan-Meier plots show the probability of hospitalization based on the length of the hospital stay, which may be misleading. Once a person stayed in the hospital, the length of stay does not affect hospitalization probability unless the authors' purpose was “the discharge probability”.

Response:

Thanks very much for reminding us. In this study, our outcome event was discharge. Survival time was hospital LoS. Survival rate means the cumulative probability of hospitalization (please see our supplement of in “Statistical analysis” part). To make it more clear, we changed “Hospitalization probability” to “Cumulative probability of hospitalization” in Figure 2 to showed the Kaplan-Meier survival curves of cumulative probability of hospitalization based on the length of the hospital stay. 

4. The authors need to describe the plots with more details in the “results of survival analysis”.

Response:

It is thoughtful of you to give us this suggestion. We added more detailed descriptions of the survival analysis results of Figure 2 and Figure 3 in the results of survival analysis section. The meaning of hazard of risk factor was added, such as “The hazard ratio of age groups indicated that compared to patients aged less than 45 years, those older than or equal to 45 years old had a 26% increase in the risk of continued hospital treatment.”

5. It would be better if the authors could provide evidence that the Cox Proportional hazard model’s assumptions are satisfied by their data.

Response:

Thanks for your suggestion. We used Schoenfeld residuals to check the proportional hazards assumption by two R packages, survival and surviviner. For each covariate, the function cox.zph() correlates the corresponding set of scaled Schoenfeld residuals with time, to test for independence between residuals and time. Additionally, it performs a global test for the model as a whole. To clarify, we added the assumptions testing method in Statistical analysis section.

Reviewer #2:

1. The title of the manuscript presents this as a prospective study. However, the study was performed on the data that was already collected through NNDRS, and filed survey. The design qualifies this as retrospective database study.

Response:

Thanks for pointing this out for us. Our data indeed was already collected through NNDRS, and filed survey. Based on the retrospective database, we changed the title “a prospective study” to “a survival analysis” and added the design qualifies as “retrospective data” in Data sources section.

2. Authors mention that all subjects gave their informed consent for inclusion before they participated. It needs further elaboration. How did subjects gave their consent if the collected data through NNDRS was obtained for the study purposes.

Response:

Thanks for pointing this out for us. We checked the ethics approval and consent to participate carefully under your suggestion. The study was approved by the Ethics Committee of Sichuan CDC (SCCDCIRB-2020-006) and the written informed consent was waived because this study, a retrospective one in nature, belongs to emergency medical service. We revised the description in this section.

3. HR below 1 intuitively mean less risk of developing an outcome/ event. Here more appropriate event should be related to Length of stay, rather than discharge. I urge authors to use HR more intuitively.

Response:

Thanks a lot for pointing this out for us. In this study, our outcome event was discharge. Survival time was hospital LoS. Survival rate means the cumulative probability of hospitalization (please see our supplement of in “Statistical analysis” part). HR<1 indicated a decreased probability of hospital discharge, which also meant a longer length of stay. To make HR more intuitively, we modified the HR description in Statistical analysis section, as follows “Briefly, hazard ratio (HR) > 1 indicated an increased probability of hospital discharge……”. We have also added more detailed descriptions of the survival analysis results of Figure 2 and Figure 3 in the results of survival analysis section. The meaning of hazard of risk factor was added, such as “The hazard ratio of age groups indicated that compared to patients aged < 45 years old, those ≥ 45 years old had a 26% increase in the risk of continued hospital treatment.”

4. The manuscript mentions that the number of confirmed cases of COVID-19 was less than 700 as of September, 2020, and more than 80% of the confirmed cases occurred before March (meaning that about 560 cases had occurred before March). Of these, the study looks at 538 hospitalized patients. Does it mean that >95% of confirmed cases were being hospitalized?

Response:

You were so kind to point this out. The figure above showed the change of daily incidence rate of novel coronavirus pneumonia in Sichuan Province in 2020 (1/10 million). The blue line represented the daily incidence rate of Sichuan Province, and the red line represented the national daily incidence rate of except Hubei Province.

In Sichuan Province, all confirmed cases were sent to hospitals for treatment. According to the development of the epidemic, we showed LoS of COVID-19 infection patients diagnosed in Sichuan from January 16, 2020 to March 4th, when almost all the cases from imported cases to local infections in Sichuan were concentrated in this period, which was the most serious epidemic period in Sichuan Province.

5. No need to mention few results. It makes the manuscript appear less thoughtful. Authors need to review which results need to be presented carefully (Under section 3.1 authors say that 271 (50%) of the 538 individuals who had to be admitted to hospital were aged at least 45 years. This is not needed because the cut-off of 45 years was chosen as it was median. Similarly, under section 3.2 no need to mention that the LoS was 19 days for both men and women.

Response:

Thank you very much for correcting us. We have checked the full text and deleted the redundant results and changed our description in section 3.1 and 3.2.

6. Surprisingly, authors do not discuss any of the multiple papers published based on data from China looking at the factors affecting LoS in COVID patients. It is important to put the finding of this paper in the context of what other researchers have found in China.

Response:

Thanks very much for your reminder. We have added the studies about risk factors related to Hospital LoS in COVID patients whose data was based on China or outside of China in the second and third paragraph of discussion section, as follows: “Hospital LoS due to COVID-19 has been reported in several studies in China. A systematic review identified 52 studies and reported that……”. 

7. The visual clarity of figures is very bad making it impossible to comment.

Response:

Thanks for pointing this out for us. We have checked all the figures’ visual clarity and focused on improving the quality of Figure 2, and added Figure S2.

---

## [Decision Letter · Decision Letter 1]

29 Nov 2021

What are the risk factors of hospital length of in the novel coronavirus pneumonia (COVID-19) patients? A survival analysis in southwest China

PONE-D-21-21225R1

Dear Dr. Zhong,

We’re pleased to inform you that your manuscript has been judged scientifically suitable for publication and will be formally accepted for publication once it meets all outstanding technical requirements.

Kind regards,

Yang Yang, Ph.D

Academic Editor

PLOS ONE

Additional Editor Comments (optional):

Reviewers' comments:

Reviewer's Responses to Questions

**Comments to the Author**

1. If the authors have adequately addressed your comments raised in a previous round of review and you feel that this manuscript is now acceptable for publication, you may indicate that here to bypass the “Comments to the Author” section, enter your conflict of interest statement in the “Confidential to Editor” section, and submit your "Accept" recommendation.

Reviewer #1: All comments have been addressed

Reviewer #2: All comments have been addressed

2. Is the manuscript technically sound, and do the data support the conclusions?

Reviewer #1: Yes

Reviewer #2: Yes

3. Has the statistical analysis been performed appropriately and rigorously? 

Reviewer #1: Yes

Reviewer #2: Yes

4. Have the authors made all data underlying the findings in their manuscript fully available?

Reviewer #1: Yes

Reviewer #2: No

5. Is the manuscript presented in an intelligible fashion and written in standard English?

Reviewer #1: Yes

Reviewer #2: Yes

6. Review Comments to the Author

Reviewer #1: All my comments and concerns are addressed by the authors. It seems the manuscript is ready to publish.

Reviewer #2: (No Response)

7. PLOS authors have the option to publish the peer review history of their article (what does this mean?). If published, this will include your full peer review and any attached files.

Reviewer #1: No

Reviewer #2: No

---

## [Editor Report · Acceptance letter]

6 Jan 2022

PONE-D-21-21225R1 

What are the risk factors of hospital length of stay in the novel coronavirus pneumonia (COVID-19) patients? A survival analysis in southwest China 

Dear Dr. Zhong:

I'm pleased to inform you that your manuscript has been deemed suitable for publication in PLOS ONE. Congratulations! Your manuscript is now with our production department. 

Kind regards, 

on behalf of

Dr. Yang Yang 

Academic Editor

PLOS ONE